# Dopamine D2 Long Receptors Are Critical for Caveolae-Mediated α-Synuclein Uptake in Cultured Dopaminergic Neurons

**DOI:** 10.3390/biomedicines9010049

**Published:** 2021-01-08

**Authors:** Ichiro Kawahata, Tomoki Sekimori, Haoyang Wang, Yanyan Wang, Toshikuni Sasaoka, Luc Bousset, Ronald Melki, Tomohiro Mizobata, Yasushi Kawata, Kohji Fukunaga

**Affiliations:** 1Department of Pharmacology, Graduate School of Pharmaceutical Sciences, Tohoku University, Sendai 980-8578, Japan; kawahata@tohoku.ac.jp (I.K.); tomoki.sekimori.q6@dc.tohoku.ac.jp (T.S.); wang.haoyang.t3@dc.tohoku.ac.jp (H.W.); 2Department of Pharmaceutical Sciences, Ben and Maytee Fisch College of Pharmacy, University of Texas at Tyler, 3900 University Blvd, Tyler, TX 75799, USA; yanyanwang@uttyler.edu; 3Department of Comparative and Experimental Medicine, Brain Research Institute, Niigata University, Niigata 951-8585, Japan; sasaoka@bri.niigata-u.ac.jp; 4Laboratory of Neurodegenerative Diseases, Institut François Jacob (MIRcen), Centre National de la Recherche Scientifique (CNRS), Commissariat à l’Énergie Atomique et aux Énergies Alternatives (CEA), 18 Route du Panorama, 92265 Fontenay-aux-Roses, France; luc.bousset@cnrs.fr (L.B.); ronald.melki@cnrs.fr (R.M.); 5Department of Chemistry and Biotechnology, Graduate School of Engineering Tottori University, Tottori 680-8550, Japan; mizobata@tottori-u.ac.jp (T.M.); kawata@tottori-u.ac.jp (Y.K.)

**Keywords:** dopamine D_2L_ receptor, fatty acid-binding protein 3 (FABP3), α-synuclein, dopaminergic neurons, synucleinopathy, Parkinson’s disease

## Abstract

α-synuclein accumulation into dopaminergic neurons is a pathological hallmark of Parkinson’s disease. We previously demonstrated that fatty acid-binding protein 3 (FABP3) is critical for α-synuclein uptake and propagation to accumulate in dopaminergic neurons. FABP3 is abundant in dopaminergic neurons and interacts with dopamine D2 receptors, specifically the long type (D_2L_). Here, we investigated the importance of dopamine D_2L_ receptors in the uptake of α-synuclein monomers and their fibrils. We employed mesencephalic neurons derived from dopamine D_2L_^−/−^, dopamine D2 receptor null (D2 null), FABP3^−/−^, and wild type C57BL6 mice, and analyzed the uptake ability of fluorescence-conjugated α-synuclein monomers and fibrils. We found that D_2L_ receptors are co-localized with FABP3. Immunocytochemistry revealed that TH^+^ D2L^−/−^ or D2 null neurons do not take up α-synuclein monomers. The deletion of α-synuclein C-terminus completely abolished the uptake to dopamine neurons. Likewise, dynasore, a dynamin inhibitor, and caveolin-1 knockdown also abolished the uptake. D_2L_ and FABP3 were also critical for α-synuclein fibrils uptake. D_2L_ and accumulated α-synuclein fibrils were well co-localized. These data indicate that dopamine D_2L_ with a caveola structure coupled with FABP3 is critical for α-synuclein uptake by dopaminergic neurons, suggesting a novel pathogenic mechanism of synucleinopathies, including Parkinson’s disease.

## 1. Introduction

The synaptic protein α-synuclein is the major component of Lewy bodies and Lewy neurites, which are the pathological hallmarks of Parkinson’s disease [1,2]. The intracellular accumulation of α-synuclein and its fibrillation increases the phosphorylation level of tyrosine hydroxylase (TH), the rate-limiting enzyme of dopamine [3]. Pathological elevation of TH phosphorylation leads to its degradation by the ubiquitin-proteasome system and loss of TH protein in dopaminergic neurons [3,4]. Additionally, phosphorylated TH has biological characteristics to readily form intracellular aggregates [5]. Consistently, Lewy bodies are positive for phosphorylated TH in Parkinson’s disease [6]. These biochemical characteristics of TH are hypothetically involved in the loss of dopaminergic neuronal function in Parkinson’s disease. Thus, elucidation of the physiological mechanism of how α-synuclein is taken up by dopaminergic neurons, accumulates to get insolubilized, and exhibits toxicity is important for maintaining the physiological function of the dopaminergic system.

The propagation of α-synuclein is important in the pathogenesis of Parkinson’s disease [7,8]. To propagate and influence, the protein needs to be taken up by the cells. We previously demonstrated that fatty acid-binding protein 3 (FABP3) is critical for the uptake of α-synuclein into dopaminergic neurons [9], thereby its nigrostriatal propagation [10]. α-synuclein is associated with FABPs as well as fatty acids and promotes protein aggregation [11,12,13]. Furthermore, FABP3 is an expected biomarker for Parkinson’s disease because it increases in the patients’ plasma [14,15,16]. Thus, the involvement of FABP3 in the etiology of Parkinson’s disease and unveiling its mechanism are anticipated to be elucidated. Unfortunately, the FABP-dependent mechanism for α-synuclein uptake into dopaminergic neurons is still not well understood.

FABPs are members of the intracellular lipid-binding protein family and are involved in reversibly binding hydrophobic ligands. FABP3 is associated with dopamine D2 receptors [17], especially with D2 long type (D_2L_) receptors [18]. The D2 receptors are localized in both the presynapse and postsynapse, which get linked to the inhibitory G-proteins and initiate their action by inhibiting adenylate cyclase. A part of this D2 receptor signaling is regulated by α-synuclein, and its C-terminal is essential for this regulation [19]. Therefore, in this study, we generated a D_2L_-specific antibody and investigated the physiological significance of the D_2L_ receptor in FABP3-dependent uptake of α-synuclein using cultured mesencephalic neurons derived from D_2L_ deficient (D_2L_^−/−^), FABP3 knockout (FABP3^−/−^), and wild type (WT) mice. We found that D_2L_ receptors are critical for caveola-mediated α-synuclein uptake into dopaminergic neurons. Moreover, α-synuclein monomers as well as fibrils are taken up into dopaminergic neurons in the D_2L_ and FABP3-dependent manner. Our findings suggest that D_2L_ receptors are important for regulating caveola-associated uptake of α-synuclein monomers as well as fibrils coupled with FABP3.

## 2. Materials and Methods

### 2.1. Animals

Pregnant C57BL/6J wild type mice were purchased from Japan SLC (Shizuoka, Japan). D_2L_^−/−^ [20], D2 null, and FABP3^−/−^ mice [21,22] were housed under climate-controlled conditions with a 12 h light and dark cycle. Primary cultured dopaminergic neurons were prepared from mice at the embryonic stage of 15.5th days. The experiments were conducted by employing two to three pregnant mice (4–12 embryos from one mother mouse) in each group and were performed in triplicates to test reproducibility. The number of animals used was the minimum number consistent with the aims of the experiment. Animal studies were conducted in accordance with the Tohoku University institutional guidelines. Ethical approval was obtained from the Institutional Animal Care and Use Committee of the Tohoku University Environmental and Safety Committee [2019PhLM0-021 (approved date: 1 December 2019) and 2019PhA-024 (approved date: 1 April 2019)].

### 2.2. Cell Culture

Primary cultures of mesencephalic neurons were prepared as previously described [4,9]. In brief, the mesencephalon was dissected from mouse embryos on the 15.5th day of gestation. The dissected tissues were then treated with papain (Sumitomo Bakelite, Tokyo, Japan) at 37 °C for 30 min and mechanically dissociated into single-cell suspensions. The cells were plated onto poly l-lysine-coated cover glass chambers at a density of 3 × 10^5^ cells/cm^2^. The cultures were maintained in Eagle’s minimum essential medium (Wako, Osaka, Japan) supplemented with 10% fetal calf serum for 1–4 days in vitro (DIV) or 10% horse serum for 5–12 DIV. The cells were cultured at 37 °C in an atmosphere with 5% CO_2_ in air and 100% relative humidity. All animal experiments were conducted in accordance with the general guidelines for animal experiments at Tohoku University.

### 2.3. Reagents

ATTO-550-labeled α-synuclein monomers and fibrils were prepared as previously described [23]. Cultured neurons were either treated or not treated with 1 μM ATTO-550-labeled α-synuclein monomers or fibrils for 48 h. To examine the dependency on caveolae in the process of α-synuclein uptake, dopaminergic neurons were exposed to 50 μM dynasore [24] (Tocris Bioscience, Bristol, United Kingdom) or 3.3 nM caveolin-1 siRNA (#sc-29942; Santa Cruz Biotechnology, Dallas, TX, USA).

### 2.4. Immunocytochemistry

Cultured mesencephalic neurons were fixed for immunocytochemistry at 12 DIV. Following fixation with 4% paraformaldehyde for 30 min, the cells were incubated with 0.1% Triton X-100 for 15 min. After pre-blocking with 5% goat serum in phosphate-buffered saline (PBS) for 1 h, they were incubated overnight at 4 °C with the following primary antibodies: rabbit anti-TH affinity-purified polyclonal antibody (1:400; Millipore, AB152, Billerica, MA, USA), mouse anti-TH monoclonal antibody (1:200; Millipore, MAB318), mouse anti-human FABP3 monoclonal antibody, clone 66E2 (1:50; Hycult Biotech, HM2016, Uden, Netherlands), rabbit anti-dopamine D2 receptor (DRD2) polyclonal antibody (1:500; Proteintech, 55084-l-AP, Rosemont, IL, USA), and rabbit anti-caveolin-1 polyclonal antibody (1:500; Abcam, ab2910, Cambridge, UK). To visualize dopamine D_2L_ receptors, we generated a D_2L_-specific rabbit polyclonal antibody (Eurofin Genomics, Tokyo, Japan). After washing with PBS, the cells were incubated with either Alexa Fluor 405-, 488-, or 546-conjugated secondary antibodies (1:500 dilution, Invitrogen, Carlsbad, CA, USA). Images were acquired using a confocal laser scanning microscope (TCS SP8, Leica Microsystems, Wetzlar, Germany). For live cell imaging, a fluorescence microscope (BZ-X810, Keyence, Osaka, Japan) was used.

### 2.5. Analyses of Fluorescence Intensity and Morphological Characteristics

Quantitative analysis of fluorescence intensity and Z-dimension analysis were performed using NIH ImageJ 1.53 software (National Institutes of Health, Bethesda, MD, USA) as previously described [5,9]. First, the background signal intensities were measured from regions without any cells and subtracted from all the images, and the remaining signals of the cells were used to define total cell areas.

### 2.6. Statistical Analyses

All values are expressed as mean ± standard error of the mean (SEM). Statistical significance was tested by one-way analysis of variance (ANOVA) with post-hoc Tukey’s multiple comparison test or two-way ANOVA with post-hoc Bonferroni’s multiple comparison test. A *p*-value < 0.05 was considered as statistically significant. All the statistical analyses were performed using GraphPad Prism 8 (GraphPad Software, San Diego, CA, USA).

## 3. Results

### 3.1. Generation of D_2L_ Specific Antibody and the Co-Localization Analysis of D_2L_ with FABP3 Distribution in Cultured Dopaminergic Neurons

To investigate the physiological significance of D_2L_ in α-synuclein uptake coupled with FABP3, we first generated an anti-D_2L_ antibody that specifically recognizes the D_2L_ specific sequence CTHPEDMKL, which does not exist in D2 short type receptors (D_2S_) (Figure 1). Immunocytochemical analysis revealed that the immunoreactivities of the generated anti-D_2L_ antibody (red) were observed in WT-derived TH^+^ neurons (green), whereas no immunoreactivities were observed in D_2L_^−/−^ TH^+^ neurons, indicating that this anti-D_2L_ antibody specifically identifies D_2L_ receptors in the mesencephalic neurons (Figure 2A). Importantly, D_2L_ (red) was well co-localized with FABP3 (green) in TH^+^ neurons (blue) in WT, whereas no D_2L_ signal was observed in D_2L_^−/−^ TH^+^ neurons (Figure 2B).

### 3.2. D_2L_ Is Critical for α-Synuclein Uptake in Cultured Dopaminergic Neurons

We investigated the physiological significance of D_2L_ in α-synuclein uptake in dopaminergic neurons. To analyze D_2L_ dependency in the uptake of α-synuclein, we exposed cultured neurons derived from either C57BL6 wild type (WT) mice, D_2L_ knockout mice (D_2L_^−/−^), or D2null mice (D2 null) to 1 μM ATTO-550-labeled α-synuclein monomer at DIV 10. We discovered that the uptake ability of ATTO-550-labeled α-synuclein monomers was drastically abolished in D_2L_^−/−^ neurons, whereas WT TH^+^ neurons took them up (Figure 3A, **** *p* < 0.0001 vs. WT). The uptake of the α-synuclein monomer significantly increased at 6 h after the exposure to 1 μM α-synuclein monomer ATTO-550 in the NeuO^+^ and FFN206^+^ neurons, markers for neuronal and VMAT2 positive cells, respectively (Appendix A). Additionally, we also investigated the ability of α-synuclein uptake in D2 null mice-derived mesencephalic neurons and revealed that the D2 receptor is critical for the uptake of ATTO-550-labeled α-synuclein monomers in TH^+^ neurons (Figure 3B, **** *p* < 0.0001 vs. WT). These data indicate that the dopamine D2 receptor, especially the D_2L_ type, is important for α-synuclein uptake into dopaminergic neurons.

### 3.3. The C-Terminal of α-Synuclein Is Essential for the Uptake into Dopaminergic Neurons

We further investigated the molecular mechanism of α-synuclein uptake in dopaminergic neurons. Since the C-terminal of α-synuclein is assumed to perform an extensive role in neurodegeneration [25,26,27,28], we analyzed the importance of α-synuclein C-terminal region for its uptake in dopaminergic neurons in our culture system. First, we generated the C-terminal-truncated α-synuclein Δ130–140 (Figure 4A). In the uptake analysis at 48 h after the addition of fluorescence-labeled WT or Δ130–140 α-synuclein, the truncated form showed a reduction in its uptake into TH^+^ neurons (Figure 4B, **** *p* < 0.0001). These data suggest that the C-terminal 131–140 amino acids of α-synuclein is important for its uptake by dopaminergic neurons.

### 3.4. α-Synuclein Uptake Is Mediated by Dynamin and Caveolin-1 in Dopaminergic Neurons

α-synuclein interacts with various membrane proteins [29] and regulates dopamine D2 receptor signaling [19]. Since dopamine D2 receptors are abundant in membrane rafts [30] and internalize via the caveolae-mediated endocytic pathway [31], we next disrupted caveola to sequester D2 receptors by dynasore, an inhibitor of dynamin-dependent endocytosis [32], or caveolin-1 knockout to examine the importance of caveola structure on the D2 receptor-dependent α-synuclein uptake in dopaminergic neurons (Figure 5A). Intriguingly, disruption of caveolae by dynasore as well as caveolin-1 knockdown abolished the uptake of α-synuclein monomers (Figure 5B, **** *p* < 0.0001). These data indicate that in addition to D_2L_ receptors themselves, caveola structures are also required for α-synuclein uptake.

### 3.5. D_2L_ Receptors and FABP3 Are Critical for the Uptake of α-Synuclein Fibrils and Monomers in Dopaminergic Neurons

The fibril form of α-synuclein shows a higher interaction with the membrane and exacerbates cytotoxicity in cultured cell lines [33]. Thus, we examined whether D_2L_ is critical for α-synuclein monomers as well as fibrils. We previously demonstrated that FABP3 is critical for the uptake of α-synuclein monomers in TH^+^ cells [9]. Therefore, we first determined that FABP3 is also required for α-synuclein fibril uptake. Importantly, TH^+^ dopaminergic neurons did not take up α-synuclein fibrils in the absence of FABP3 (Figure 6A, FABP3^−/−^; Figure 6B, *p* < 0.0001 vs. WT). Moreover, D_2L_ knockout showed a prominent decrease in the ability of α-synuclein fibril uptake (Figure 6A, D_2L_^−/−^; Figure 6B *p* < 0.0001 vs. WT). In the TH^+^ neurons derived from WT mice, α-synuclein fibrils formed intracellular aggregations in the cell body (Figure 6A, WT) and showed Lewy neurite-like morphologies in the neuronal processes (Figure 6C). These data suggest that both FABP3 and D_2L_ receptors are critical for the α-synuclein fibrils as well as monomers.

### 3.6. D_2L_ Receptors Are Predominantly Co-Localized with α-Synuclein Fibril ATTO-550 in Dopaminergic Neurons

Finally, we investigated the spatial characteristics of D_2L_ receptors coupled with α-synuclein fibril uptake. After 48 h of the treatment with 1 μM α-synuclein fibril ATTO-550, cells were fixed and analyzed through immunocytochemistry. Importantly, we found that α-synuclein fibrils taken up by TH^+^ neurons were eminently co-localized with D_2L_ immunoreactivities (Figure 7A, WT). Z-dimension analysis revealed that D_2L_ tightly co-localized with α-synuclein fibrils (Figure 7B). D_2L_^−/−^ TH^+^ neurons showed no intracellular aggregation of α-synuclein fibrils (Figure 7A, D_2L_^−/−^). These data indicate that D_2L_ receptors are involved in the uptake process of α-synuclein fibrils and their intracellular aggregation.

## 4. Discussion

In the current study, we demonstrated that dopamine D_2L_ receptor is critical for the uptake of α-synuclein fibrils and monomers in dopaminergic neurons using D_2L_-deficient mice, specifically lacking dopamine D2 long type receptors, as well as D2 null mice. We also employed FABP3^−/−^ mice and found that FABP3 is critical for the uptake of α-synuclein fibrils in addition to monomers [9]. D_2L_ receptors and FABP3 (Figure 2) as well as D_2L_ and α-synuclein fibrils (Figure 7) were highly co-localized. Dopaminergic neurons required the C-terminal region of α-synuclein to take up the protein into the cells (Figure 4). Additionally, dopaminergic neurons took up α-synuclein in a caveola-dependent manner (Figure 5). Our data indicate a novel mechanism by which D_2L_ receptors are required for the caveola-mediated α-synuclein uptake through its C-terminal region coupled with FABP3 (Figure 8).

Caveola is a lipid raft invagination of the plasma membrane with a defined omega-shaped structure [34,35,36]. Caveolae are principally composed of caveolin proteins (Figure 5 and Figure 8). There are three caveolin proteins, caveolin-1, caveolin-2, and caveolin-3 [34]. In neuronal cells, caveolae formation is strictly dependent on caveolin-1 [34], which was knocked down in this study to elucidate the caveola-dependent uptake of α-synuclein. Caveolin-1 interacts with α-synuclein [37], suggesting that the uptake of the protein might be mediated by caveolae-mediated endocytosis. The internalization of caveolae is also mediated by dynamin in mammalian cells [38]. Consistent with these results, we demonstrated that caveolin-1 knockdown or treatment with dynasore, a cell-permeable inhibitor of dynamin [32], abolishes α-synuclein uptake into dopaminergic neurons (Figure 5). Dynamin-dependent α-synuclein uptake is also modulated by clusterin in the case of astrocytes [39], indicating the possible mechanism in clusterin-mediated regulation of neuronal uptake inhibition of extracellular α-synuclein.

Dopamine D_2L_ receptors bind to FABP3 by the 29 amino acid domain of the third cytoplasmic loop [18,40,41]. Furthermore, FABP3 is localized in mature neurons and interacts with α-synuclein in dopaminergic neurons in the substantia nigra [13,17,42,43], and directly binds to α-synuclein in vitro (in submission). Additionally, α-synuclein itself has the ability to interact with the plasma membrane to pass through by the formation of short fibrils [33]. In this study, we provided evidence that D_2L_ receptors and FABP3 are critical for α-synuclein uptake coupled with caveolae formation in TH^+^ neurons. These data suggest a possible mechanism by which dopaminergic neurons passively take up membrane-bound α-synuclein in a caveolae-mediated and D_2L_-FABP3-associated manner (Figure 8).

Dopamine D_2S_ and D_2L_ receptors may differentially contribute to dopaminergic signaling [44,45,46]. Interestingly, deletion of D_2L_ diminishes the typical antipsychotic raclopride-induced parkinsonism [44]. These data suggest the potential of drug development targeting the D_2L_ receptors to sequester α-synuclein to prevent its uptake into dopaminergic neurons in Parkinson’s disease therapy. D_2L_ receptors play a more prominent role than D_2S_ in mediating emotional response, such as behavioral reactions to novelty and inescapable stress [47], indicating the possibility that protection of D_2L_ from α-synuclein is important in the uptake and propagation process of the protein. In addition to the FABP ligands [48,49,50], we will try to develop the molecules targeting α-synuclein to prevent its uptake into dopaminergic cells.

In conclusion, the present study demonstrated the physiological significance of dopamine D_2L_ receptors in the uptake and accumulation process of α-synuclein in neuronal cells. D_2L_ was critical for α-synuclein uptake coupled with FABP3 in a caveola-dependent manner. In addition, D_2L_ and FABP3 are crucial for the uptake of α-synuclein fibrils as well as monomers. Our data suggest that D_2L_ is a potential target for the development of prophylactic medicine by preventing α-synuclein propagation in the neurodegenerative process in Parkinson’s disease and other synucleinopathies.

## Figures and Tables

**Figure 1 biomedicines-09-00049-f001:**
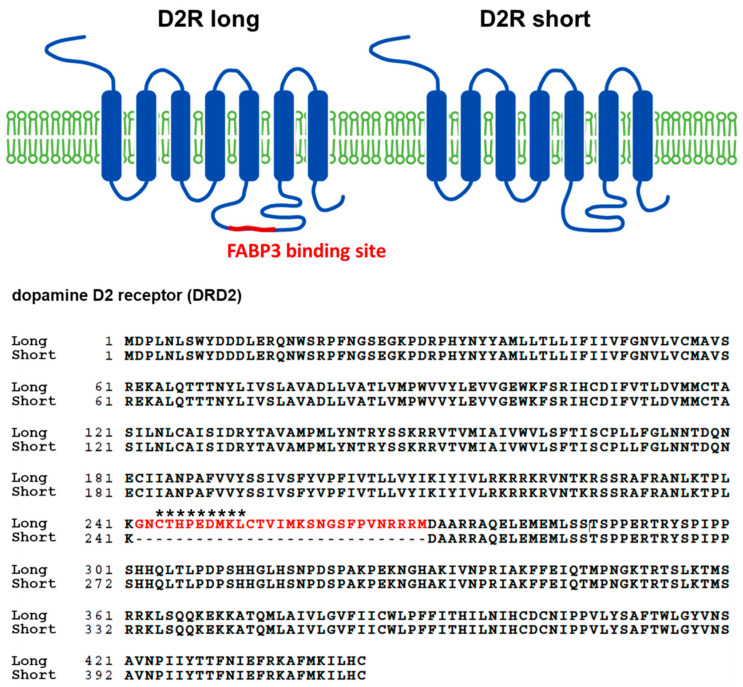
The primary sequence of mouse dopamine D2 receptor long (**left**) and short (**right**) isoforms. Amino acid sequences in red letters are specifically expressed in the long type. The dopamine D2 receptor short isoform (**right**) lacks these 29 amino acid sequences, which interact with fatty acid-binding protein 3 (FABP3). Asterisks indicate the antigen used to produce D_2L_ specific antibody used in this study.

**Figure 2 biomedicines-09-00049-f002:**
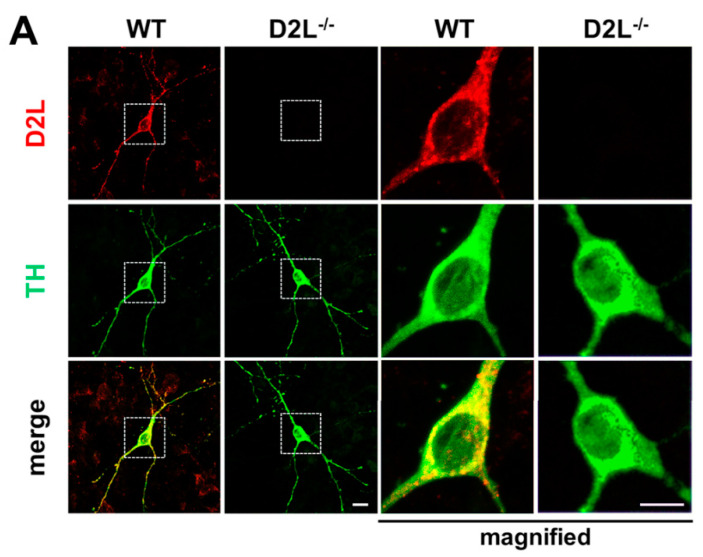
Distribution of dopamine D2 long (D_2L_) receptors and their co-localization with FABP3 in dopaminergic neurons. (**A**) Cultured mesencephalic neurons were stained with anti-D_2L_-specific (red) and anti-tyrosine hydroxylase) (TH (green) antibodies. No D_2L_ signal was observed in D_2L_^−/−^ neurons. Right images in each group were magnified by three times. Scale bar 10 μm. (**B**) Cultured mesencephalic neurons were counterstained with anti-D_2L_-specific (red), anti-FABP3 (green), and anti-TH (red) antibodies. D_2L_ and FABP3 were co-distributed. Right images in each group were magnified by 2.5 times. Scale bar 10 μm.

**Figure 3 biomedicines-09-00049-f003:**
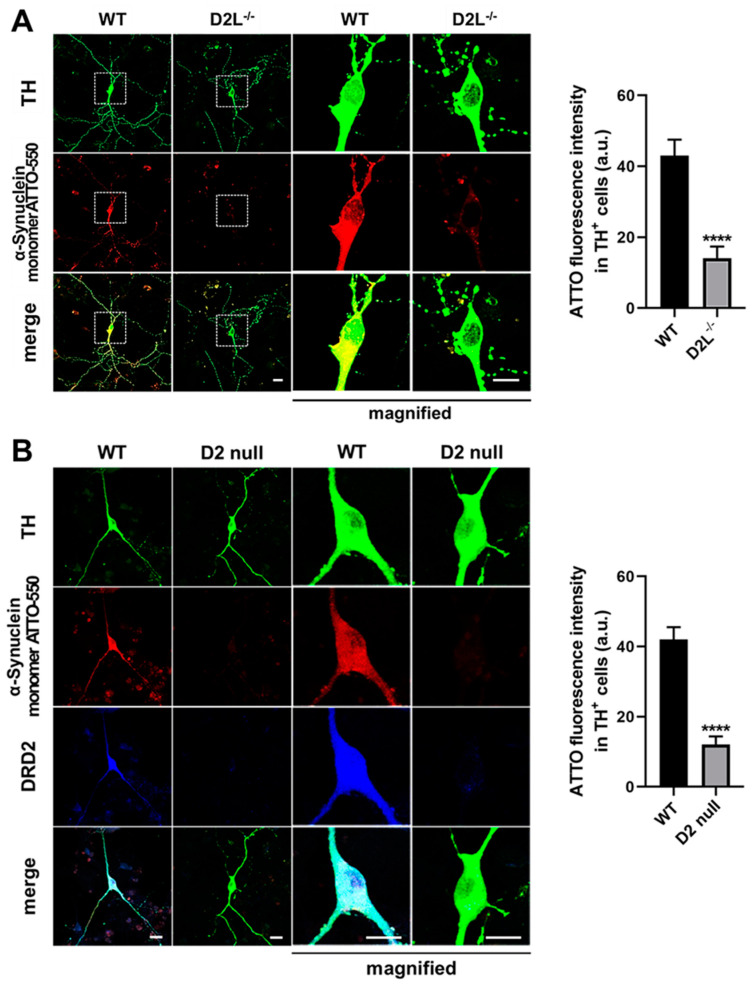
Cultured dopaminergic neurons require dopamine D2 receptors to take up α-synuclein. (**A**) Representative images of TH^+^ mesencephalic neurons at 12 days in vitro (DIV) derived from wild type (WT) or D_2L_^−/−^ C57BL6 mice. Neurons were treated with 1 μM ATTO-550-labeled α-synuclein monomer for 48 h and stained with anti-TH antibody (TH, green). The magnified images were enlarged by three times. Scale bar 10 μm. The right graph shows the quantitative analysis of ATTO-550-labeled α-synuclein monomer fluorescence intensity of individual TH^+^ neurons. **** *p* < 0.0001 in wild type (WT) versus D_2L_^−/−^, n = 34 in three independent experiments. (**B**) Representative images of TH^+^ mesencephalic neurons derived from wild type or D2 null knockout mice. Neurons were treated with ATTO-550-labeled α-synuclein monomer in the same condition as in (**A**) and stained with anti-TH antibody (TH, green) and dopamine D2 receptor (DRD2, blue). The magnified images were enlarged by three times. Scale bar 10 μm. The quantitative analysis of ATTO-550-labeled α-synuclein monomer fluorescence intensity of individual TH^+^ neurons on the right. **** *p* < 0.0001 in WT versus D2 null knockout (D2 null), n = 28 in three independent experiments.

**Figure 4 biomedicines-09-00049-f004:**
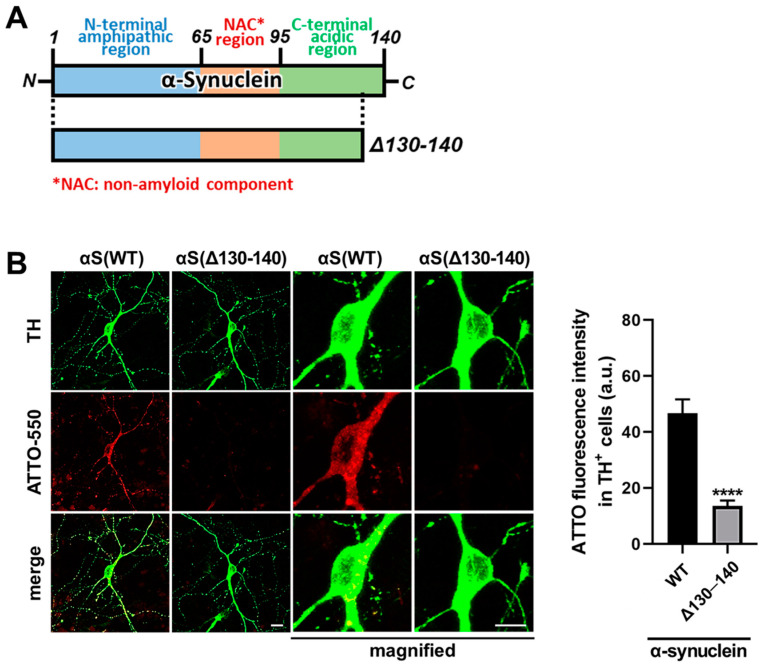
Structure of α-synuclein mutant lacking C-terminal region and the effect of C-terminus deletion on the ability of α-synuclein uptake in dopaminergic neurons. (**A**) The structure of a mutant form of α-synuclein with an 11-amino acid deletion in the C-terminal region. (**B**) Cultured mesencephalic neurons were treated with 1 μM ATTO-550-labeled wild type or Δ130–140 deletant α-synuclein monomer for 48 h. C-terminus deletion decreased the ability of α-synuclein uptake. The magnified images were enlarged by three times. Scale bar 10 μm. The right graph shows the quantitative analysis of ATTO-550-labeled α-synuclein monomer fluorescence intensity of individual TH^+^ neurons. **** *p* < 0.0001 in α-synuclein wild type (WT) versus Δ130–140 deletant (Δ130–140), n = 27 in three independent experiments.

**Figure 5 biomedicines-09-00049-f005:**
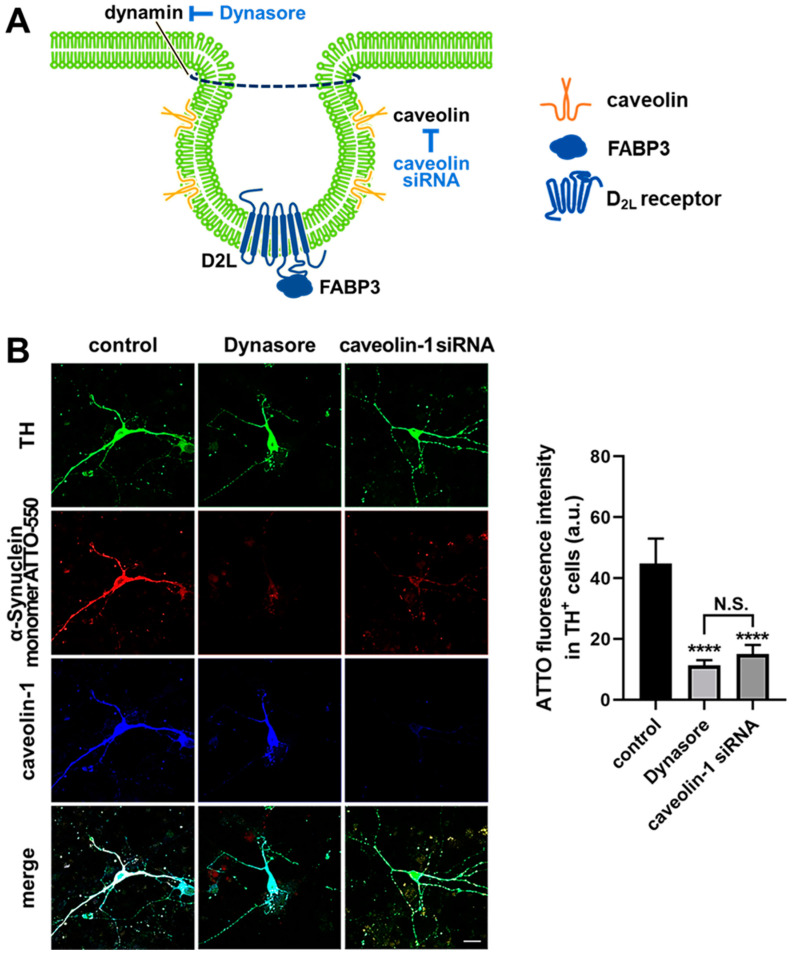
Schematic illustration for the inhibition of caveolae formation and the effect of the inhibition on the α-synuclein uptake in dopaminergic neurons. (**A**) Schematic illustration for the inhibition of caveolae formation by a dynamin inhibitor dynasore and caveolin knockdown by siRNA. D_2L_ receptors are abundant in the caveolae. (**B**) Effect of the treatment with dynasore and caveolin siRNA on the ability of α-synuclein uptake in dopaminergic neurons. Cells were stained with anti-TH (green) and anti-caveolin-1 (blue) antibodies. Scale bar 10 μm. The quantitative analysis of ATTO-550-labeled α-synuclein monomer fluorescence intensity of individual TH^+^ neurons on the right. **** *p* < 0.0001 in control versus dynasore and caveolin-1 siRNA, n = 33 in three independent experiments. N.S. means no significant difference was observed between dynasore and caveolin-1 knockdown.

**Figure 6 biomedicines-09-00049-f006:**
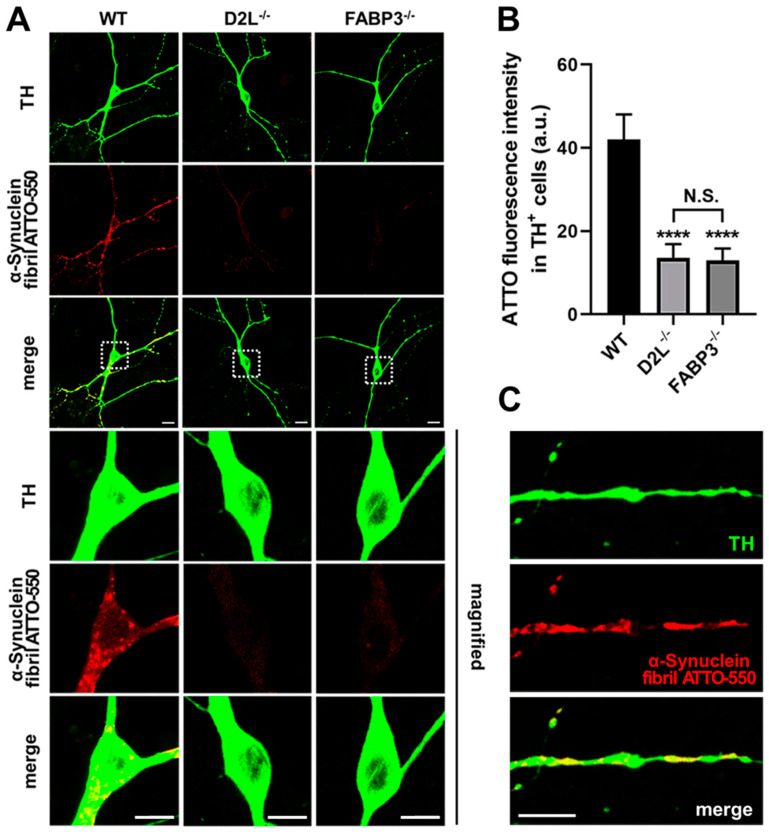
Cultured dopaminergic neurons require dopamine FABP3 and D_2L_ receptors to take up α-synuclein fibrils. (**A**) Representative images of TH^+^ mesencephalic neurons derived from wild type (WT), D_2L_^−/−^ or FABP3^−/−^ C57BL6 mice. Neurons were treated with 1 μM ATTO-550-labeled α-synuclein fibrils for 48 h and stained with anti-TH antibody (TH, green). The magnified images were enlarged by three times. Scale bar 10 μm. (**B**) The quantitative analysis of ATTO-550-labeled α-synuclein fibril fluorescence intensity of individual TH^+^ neurons. **** *p* < 0.0001 versus WT, n = 31 in three independent experiments. N.S. means no significant difference was observed between D_2L_^−/−^ and FABP3^−/−^. (**C**) Lewy neurites-like morphology of the accumulated α-synuclein fibrils in the neurites of dopaminergic neurons. Scale bar 10 μm.

**Figure 7 biomedicines-09-00049-f007:**
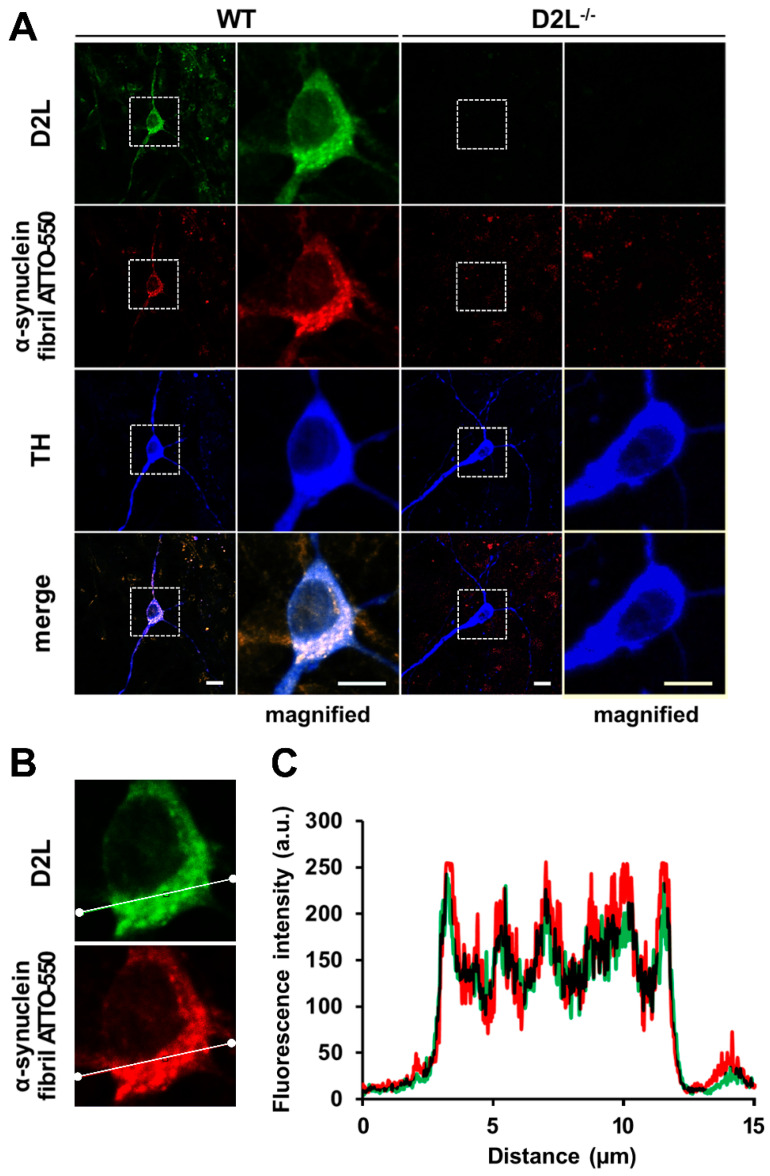
Co-localization of D_2L_ receptors with α-synuclein fibrils in dopaminergic neurons. (**A**) Cultured mesencephalic neurons were treated with 1 μM α-synuclein fibril ATTO-550 and stained with anti-D_2L_ (green) and anti-TH (blue) antibodies. The magnified images were enlarged by three times. Scale bar 10 µm. (**B**) Z-dimension analysis of D_2L_ receptor-positive dotted immunoreactivities and α-synuclein fibrils. (**C**) Quantified fluorescence intensities of D_2L_ receptors and α-synuclein fibrils analyzed in (**B**). Both signals were tightly co-distributed.

**Figure 8 biomedicines-09-00049-f008:**
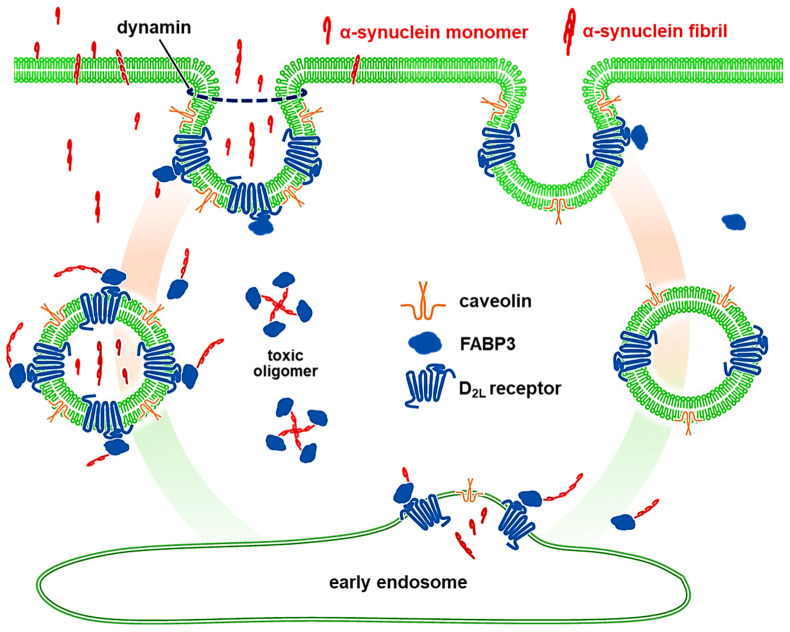
Schematic illustration of α-synuclein uptake in dopaminergic neurons. We demonstrated that dopaminergic neurons require dopamine D_2L_ receptors, FABP3, and dynamin/caveolin-1-coupled caveolae formation to take up α-synuclein monomers as well as fibrils. In this context, FABP3 and caveolin can interact with α-synuclein, and dopamine D2L receptors bind to FABP3. Caveolin and dopamine D_2L_ receptors are abundant in lipid raft. The caveolae-mediated endocytosis is coupled with dopamine D_2L_ receptors and FABP3, and the structure is abundantly associated with α-synuclein. The structure of D_2L_/FABP3 in the endocytotic process holds α-synuclein and does not release until its recycling. We also suggest that α-synuclein uptake by different mechanisms in other cell types, such as glial cells, is also conceivable.

## Data Availability

The data presented in this study are available on request from the corresponding author.

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
