# Peer review of "Dopamine D2 Long Receptors Are Critical for Caveolae-Mediated α-Synuclein Uptake in Cultured Dopaminergic Neurons"

_biomedicines, 2021, doi:10.3390/biomedicines9010049_

Round 1
Reviewer 1 Report
"Dopamine D2 long receptors are critical for caveolae- 2 mediated α-synuclein uptake in cultured 3 dopaminergic neurons."
The authors presented the results of studies on the participation of the dopamine D2 (Long type) receptor in the uptake and accumulation of synuclein in Parkinson's disease. The research method used is well chosen, the results are well illustrated and described. The authors investigated neurobiochemical changes on the animal model. The reproduced results are consistent, well-described. The data presentation is clear. They can make a significant contribution to therapy of Parkinson's disease.
To fully appreciate the authors' work, they should complete the missing data:
Minor revision:
In section 2. Materials and Methods (2.1. Animals): information on the age and number of animals used in the study is missing. There is also no short information on the method of preparing cell cultures (in section 2.2. there is only a reference to a publication not always available).
The statistics data are incomplete, the value of n (placed in the descriptions of figures) is defined as n>20 (What does it mean? How many analyzes have been performed have been performed to obtain consistent results?)
The work is very interesting, however, the authors must pay special attention to inconsistencies and errors before publication.
Author Response
Response to Reviewer 1
The authors presented the results of studies on the participation of the dopamine D2 (Long type) receptor in the uptake and accumulation of synuclein in Parkinson's disease. The research method used is well chosen, the results are well illustrated and described. The authors investigated neurobiochemical changes on the animal model. The reproduced results are consistent, well-described. The data presentation is clear. They can make a significant contribution to therapy of Parkinson's disease.
To fully appreciate the authors' work, they should complete the missing data:
Minor revision:
- In section 2. Materials and Methods (2.1. Animals): information on the age and number of animals used in the study is missing. There is also no short information on the method of preparing cell cultures (in section 2.2. there is only a reference to a publication not always available).
Ans: According to the comment, we added the description for the information on the age and number of animals used in the study to the "Materials and Methods" section as followed.
Line 76 - 80: Primary cultured dopaminergic neurons were prepared from mice at the embryonic stage of 15.5th days. The experiments were conducted by employing two to three pregnant mice (4-12 embryos from one mother mouse) in each group and were performed in triplicates to test reproducibility. The number of animals used was the minimum number consistent with the aims of the experiment.
We also added short information on the method of preparing cell cultures.
Line 85 - 88: In brief, the mesencephalon was dissected from mouse embryos on the 15.5th day of gestation. The dissected tissues were then treated with papain (Sumitomo Bakelite, Japan) at 37°C for 30 min and mechanically dissociated into single-cell suspensions.
- The statistics data are incomplete, the value of n (placed in the descriptions of figures) is defined as n>20 (What does it mean? How many analyzes have been performed have been performed to obtain consistent results?)
The work is very interesting; however, the authors must pay special attention to inconsistencies and errors before publication.
Ans: Thank you for encouragement foe our study. To clarify the statistics data, we added an accurate description of the value of number to each figure legend.
Line 168: n = 34 in three independent experiments
Line 174: n = 28 in three independent experiments
Line 192: n = 27 in three independent experiments
Line 211: n = 33 in three independent experiments
Line 232: n = 31 in three independent experiments
Line 475: n = 21 in three independent experiments
Reviewer 2 Report
The paper entitled” Dopamine D2 long receptors are critical for caveolae mediated α-synuclein uptake in cultured dopaminergic neurons”
The authors have shown that D2L receptors are critical for caveola-mediated α-synuclein uptake into dopaminergic neurons derived from cultured mouse mesencephalic neurons. Also, α-synuclein monomers as well as fibrils are taken up into dopaminergic neurons in the D2L and FABP3-dependent manner. In my opinion researchers in this field will benefit from this study and may push further studies to develop novel strategies regarding how the uptake of α-synuclein monomers as well as fibrils into dopaminergic neurons can be therapeutically restored.
The manuscript is well structured and the hypothesis was thoroughly tested. I want to congratulate the authors on their work and recommend the acceptance of the paper in Biomedicines.
Minor comments
Results
Add a legend to figure 2/B (is missed)
Discussion
Recent reports have suggested that the clearance of extracellular α‐synuclein toxic forms is crucial to control the propagation and the progression of Parkinson's disease (see Alice Filippini et al., 2020; https://doi.org/10.1002/glia.23920). Please address this issue in case of neuronal uptake inhibition of α‐synuclein.
Author Response
Response to Reviewer 2
The authors have shown that D2L receptors are critical for caveola-mediated α-synuclein uptake into dopaminergic neurons derived from cultured mouse mesencephalic neurons. Also, α-synuclein monomers as well as fibrils are taken up into dopaminergic neurons in the D2L and FABP3-dependent manner. In my opinion researchers in this field will benefit from this study and may push further studies to develop novel strategies regarding how the uptake of α-synuclein monomers as well as fibrils into dopaminergic neurons can be therapeutically restored. The manuscript is well structured and the hypothesis was thoroughly tested. I want to congratulate the authors on their work and recommend the acceptance of the paper in Biomedicines.
Minor comments
- Add a legend to figure 2/B (is missed)
Ans: We added the description for Figure 2B as followed.
Line 145 - 147: (B) Cultured mesencephalic neurons were counterstained with anti-D2L-specific (red), anti-FABP3 (green), and anti-TH (red) antibodies. D2L and FABP3 were co-distributed. Scale bar 10 μm.
- Recent reports have suggested that the clearance of extracellular α‐synuclein toxic forms is crucial to control the propagation and the progression of Parkinson's disease (see Alice Filippini et al., 2020; https://doi.org/10.1002/glia.23920). Please address this issue in case of neuronal uptake inhibition of α‐synuclein.
Ans: We added a description of the clusterin-mediated neuronal uptake inhibition of α‐synuclein as followed.
Line 281 - 283: Dynamin-dependent α-synuclein uptake is also modulated by clusterin in the case of astrocytes [39], indicating the possible mechanism in clusterin-mediated regulation of neuronal uptake inhibition of extracellular α-synuclein.